# Research on data imbalance in intrusion detection using CGAN

**Guangyu Zhao[1], Peng Liu[1]\*, Ke Sun[2], Yang Yang[1], Tianyu Lan[1], Han Yang[3]**

**1** School of Electronic Information Engineering, Changchun University of Science and Technology, Changchun, China, **2** North Navigation Control Technology CO., LTD, Beijing, China, **3** Beijing Engineering Research Center of Emergency Survival Security, Beijing, China

\* animal20221024@126.com

**Data Availability Statement:** All files can be found from the following dataset https://www.unb.ca/cic/datasets/nsl.html -NSL_KDDdatasheet https://research.unsw.edu.au/projects/unsw-nb15-dataset -UNSW_NB15datasheet.

## Abstract

To address the problems of attack category omission and poor generalization ability of traditional Intrusion Detection System (IDS) when processing unbalanced input data, an intrusion detection strategy based on conditional Generative Adversarial Networks (cGAN) is proposed. The cGAN generates attack samples that approximately obey the distribution pattern of input data and are randomly distributed within a certain bounded interval, which can avoid the redundancy caused by mechanical data widening. The experimental results show that the strategy has better performance indexes and stronger generalization ability in overall performance, which can solve insufficient classification performance and detection omission caused by unbalanced distribution of data categories and quantities.

## 1. Introduction

As social behaviors become more intelligent, the network security boundary is increasingly blurred, and the attack methods and tools for network intrusion become more and more diverse. The effective protection and safe circulation of data have become key issues for the development of digital society. Among them, data imbalance processing is an important part of network intrusion attack detection that cannot be ignored. The real network attacks include many categories of attack behavior, but they seldom happen. The proportion of various types of attack data to the total traffic data is less than 0.1%. In general, the number of attacks is small. Consequently, many intrusion detection methods in the identification process cannot learn the complete data features, or even omit them.

In recent years, the outstanding feature-extraction capability of deep learning has attracted the research interest of many scholars in the field of intrusion detection. Zhang Y et al. [1] proposed a network intrusion detection method based on autoencoder and long- and short-term memory neural network to address the problems of high data dimensionality and complex feature extraction process of traditional network intrusion detection methods. Zavrak S et al. [2], on the other hand, focused on detecting anomalous network traffic from network stream-based data using unsupervised and semi-supervised deep learning methods. Staudemeyer R C et al. [3] modeled network traffic as time series with supervised

**Funding:** This research was funded by the National Natural Science Foundation of China, grant number U2141231. the Science and Technology Development Program of Jilin Province, grant number 20200401066GX. The Science and Technology Development Plan Project of Jilin Province, grant number 20200404216YY.

**Competing interests:** The authors have declared that no competing interests exist.

learning methods, using known normal and malicious behavior data for training and improving intrusion detection. Dai Yuanfei et al. [4] investigated the problem of degradation of detection model accuracy and long training time caused by redundancy or noise in existing intrusion detection algorithms, introduced feature selection algorithms into the field of intrusion detection, and proposed a feature selection-based intrusion detection method. Yin C L et al. [5] explored how to model intrusion detection systems based on deep learning, and proposed a method of deep learning using recurrent neural network for intrusion detection. However, in the above study, with class and number imbalance, the training effect was not informative and the result was more biased towards the majority class classification result, which led to a lower final attack recognition effect than expected and a larger bias in the detection result.

In imbalanced scenarios, not only do the class and quantity imbalance ratios change over time, but the relationships between classes also change. Moreover, the cost of misclassifying abnormal behavior data as normal behavior ones is usually higher than the cost of reversing the error. To address such problems, Chawla N V et al. [6] proposed a method for constructing classifiers from unbalanced datasets, Synthetic Minority Over-sampling Technique (SMOTE). Combining minority class data oversampling and majority class data undersampling in the curve space with receiver-operated feature yielded a classifier with better performance than the method that undersamples only the majority class. To further improve the metrics, Hui H et al. [7] proposed two new minority class oversampling methods, BorderlineSMOTE1 and BorderlineSMOTE2, based on the SMOTE method. As such, only the minority class data near the boundary were oversampled. With this design, better recall and F-values were obtained. He H et al. [8] proposed a novel adaptive synthetic sampling approach for learning from unbalanced datasets. Yang Y et al. [9] proposed a novel network intrusion detection model with regularized supervised adversarial variational autoencoder, Adaptive Synthetic Sampling Approach for Imbalanced Learning (ADASYN). However, some critical issues still remained.

1. The class of the current network attack behavior is imbalanced. The existing methods are not effective enough to deal with such problem, resulting in missing classes, key information, and attacks, etc. The detection rate is ineffective.

2. The incremental data obtained by the existing oversampling methods possess fewer real features, rigid distribution patterns, excessive information redundancy, and omission of data distribution details, which are far from the real network attack behavior data.

3. The features of a few classes of data are overlapped, so the classification tend to produce data annexation. The recognition results of existing classification methods are seriously biased towards the majority class classification results, and new attack behavior data cannot be identified when they appear, so it is difficult to distinguish the distribution patterns with fluctuations in a certain range.

To solve the above problems, we propose a network intrusion detection strategy based on conditional generative adversarial network, which takes real data as the ideal learning object, randomly simulates the feature distribution of minority class data in a certain bounded interval by generative adversarial network, and approximates the real data distribution through adversarial training to obtain a large amount of selectable training data and achieve minority class data enhancement. In this way, this paper solves the insufficient classification performance and detection omission caused by the unbalanced distribution of data categories and quantities in the classification problem.

## 2. Related work

### 2.1. Generative adversarial networks

Generative Adversarial Networks (GANs) [10] is a deep generative model that consists of two competing neural network models: The Generator (G) and the Discriminator (D). The GAN network architecture is shown in Fig 1.

The continuous adversarial training of model G and model D maximizes the probability of D to discriminate the source of training samples, and maximizes the similarity between the generated data generated by G and the real data. The training of D and G can be expressed as a two-sided game problem about the minimization and maximization of the value function, that is, the loss function of GAN, which is shown in Eq (1).

$$\min_{G} \ \max_{D} \ V(D, G) = E_{x \sim P_{data}(x)}[\lg D(x)] + E_{z \sim P_z(z)}[\lg(1 - D(G(z)))] \tag{1}$$

Goodfellow et al. [10] proved that there is an overall optimal solution to the two-sided game problem with minimal maximization when and only when $p_g = p_{data}$, i.e., a Nash equilibrium is reached. At this point, the generative model G learns the distribution of the real samples, so that the accuracy of the discriminative model D stays stably above 1/2, even if D can only make random guesses between 0 or 1 for the training samples. The parameters are updated by error gradient back propagation.

### 2.2. Conditional generative adversarial networks

The cGAN [11] is a variant of generative adversarial network. The input to the original generative adversarial network generator is a random noise signal, and the input to the discriminator is real data and generated data. The input to the conditional generative adversarial network

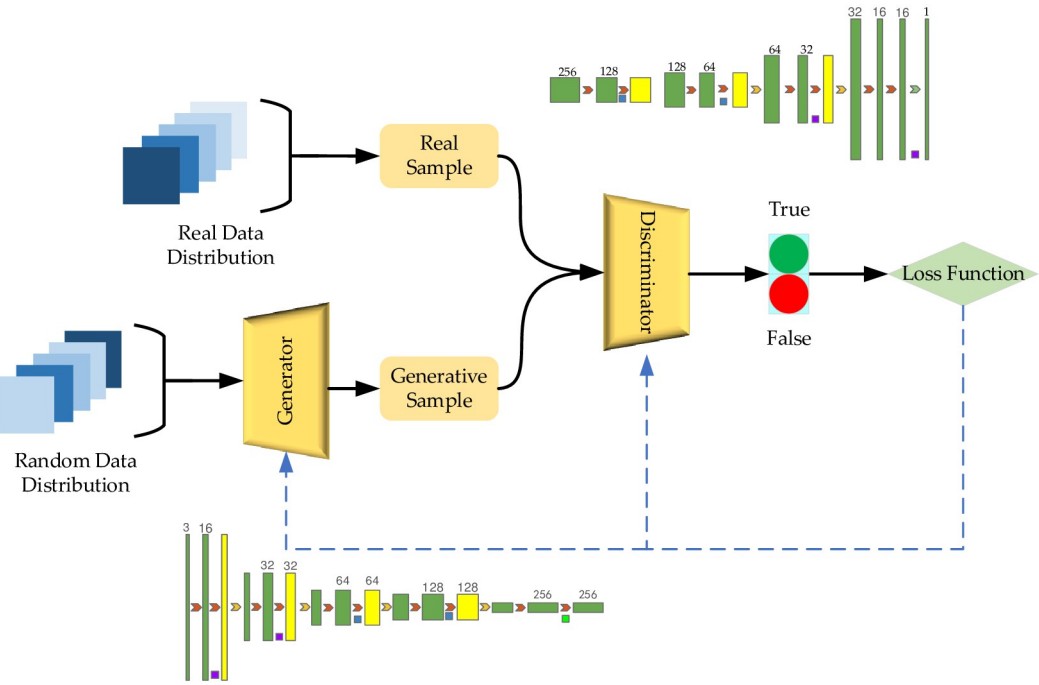

**Fig 1. GAN network architecture.**

generator is a combination of signals consisting of conditional information and random noise, and the input to the discriminator is the reconstructed data after splicing the conditional information from the real and generated data respectively. The training process is similar to that of the original generative adversarial network so it is not described here. The loss function of the conditional generative adversarial network is shown in Eq (2).

$$\min_{G} \ \max_{D} \ V(D, G) = \mathrm{E}_{x \sim P_{data}(x)}[lg \, D(x|c)] + \mathrm{E}_{z \sim P_z(z)}[lg(1 - D(G(z|c)))] \tag{2}$$

## 2.3. Intrusion detection

Intrusion detection is the discovery of intrusion behavior, an attempt to detect intrusion by observing behavior, security logs, or auditing data. The software and hardware for intrusion detection form the intrusion detection system. It collects and analyzes information from key points in a computer network or computer system from which it discovers and reacts to violations of security policies and signs of being attacked. The Common Intrusion Detection Framework (CIDF) [12] proposed by the Defense Advanced Research Projects Agency (DARPA) describes a common model for intrusion detection, as shown in Fig 2.

With the development of the network, the technological methods of fusion anomaly detection and misuse detection models are gradually formed, as shown in Fig 3. The captured network data is preprocessed, and data in the application layer are whitelisted through the whitelist database. If the matching is successful, the data in the application layer and the instruction data are put into the state machine for execution, and then the state analyses are performed. If not, the system collects status statistics directly. Finally, analyses and corresponding processing are carried out. At the same time, the abnormal patterns that cannot be associated should be added to the abnormal data set and fed back to the blacklist database and whitelist database for update through data mining.

# 3. Strategic design

## 3.1. Analyses of imbalanced data distribution

NSL-KDD [13] is a dataset proposed to solve the inherent problems in the KDD99 dataset. Many studies use it as an effective benchmark dataset to help researchers compare different intrusion detection methods. Therefore, the evaluation results of different research efforts based on the NSL-KDD dataset are consistent and comparable.

The NSL-KDD dataset includes five types of data, including normal traffic data and four major categories of attack behavior data. Each behavioral information can be divided into 43

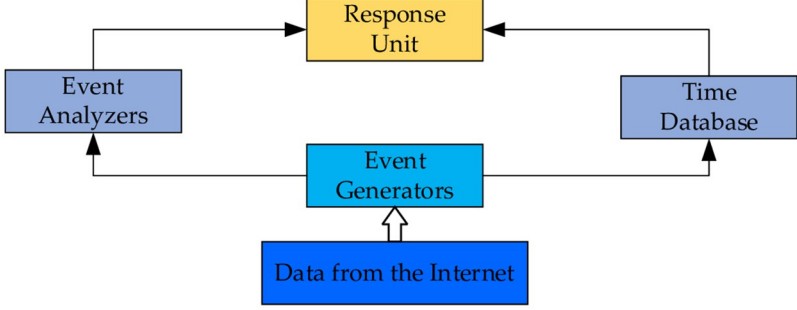

**Fig 2. General model of intrusion detection.**

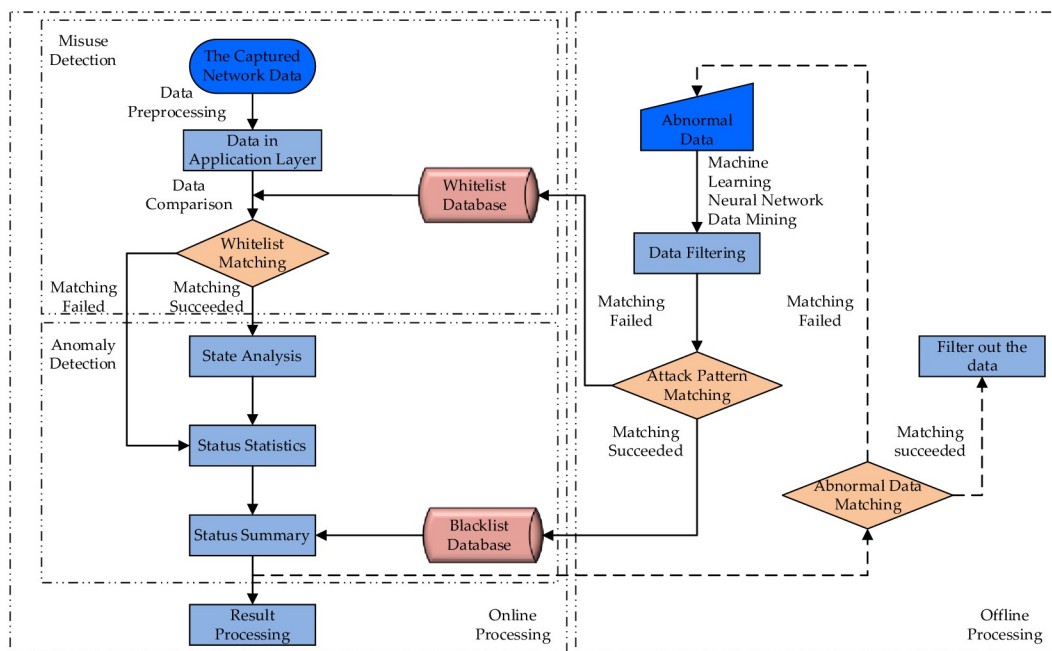

**Fig 3. Technological methods of fusion anomaly detection and misuse detection.**

dimensions according to data dimensions, among which from column 1 to column 41 are the characteristics of the network data flow itself. Column 42 is the category label, the irrelevant data for this experiment that need to be removed during data processing.

The analyses of the attack samples are shown in Fig 4. And combined with the proportion analysis of Fig 5, the distribution of categories is extremely unbalanced, with 12 categories accounting for about 0.1%, which are guess_passwd(0.042%), buffer_overflow(0.024%), warez-master(0.016%), land(0.014%), imap(0.009%), rootkit(0.008%), loadmodule(0.007%), ftp_write(0.006%), multihop(0.006%), phf(0.003%), perl(0.002%) and spy(0.002%). The percentage of each type of data varies greatly. The number of unbalanced classes accounts for 52.17% of the total number of classes, and the amount of data only accounts for 1.39% of the total amount of data, which tend to mislead the classifier and cause detection omission and inaccurate identification. In addition, since the percentage is small, even if the abnormal behavior is recognized, the recognition rate is too low to attract attention.

To further verify the effect of unevenly distributed data on the classifier, the decision tree model is used to classify the features of the KDDTrain+.txt file. The classification results are shown in Table 1 with key information highlighted. It can be seen that the class distribution is extremely uneven, resulting in unsatisfactory categories of accuracy, recall, F-value and support, or even being 0, which does not have any reference value. The detection results do not show phf category attack behavior, which is a missed detection behavior, meaning a reduced reference value and effectiveness of the classification accuracy. In addition, in the multi-classification results of these 23 types of data, the macroavg value of precision is 0.72, the macroavg value of recall is 0.75, and the macroavg value of F-value is 0.72. It can be seen that the result is very unsatisfactory. In real network intrusion attacks, the distribution of attack behavior compared to normal behavior and this situation is similar. Therefore, great attention is needed to the problem of unbalanced data distribution, which is also the focus of this paper.

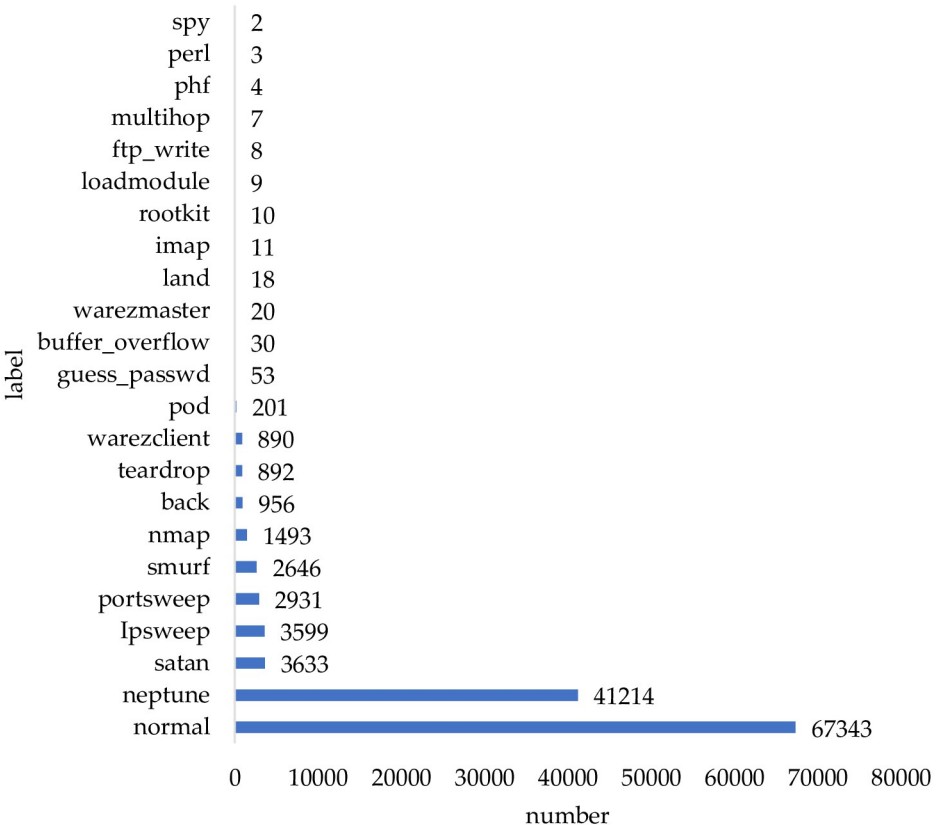

**Fig 4. Distribution of the original label data.**

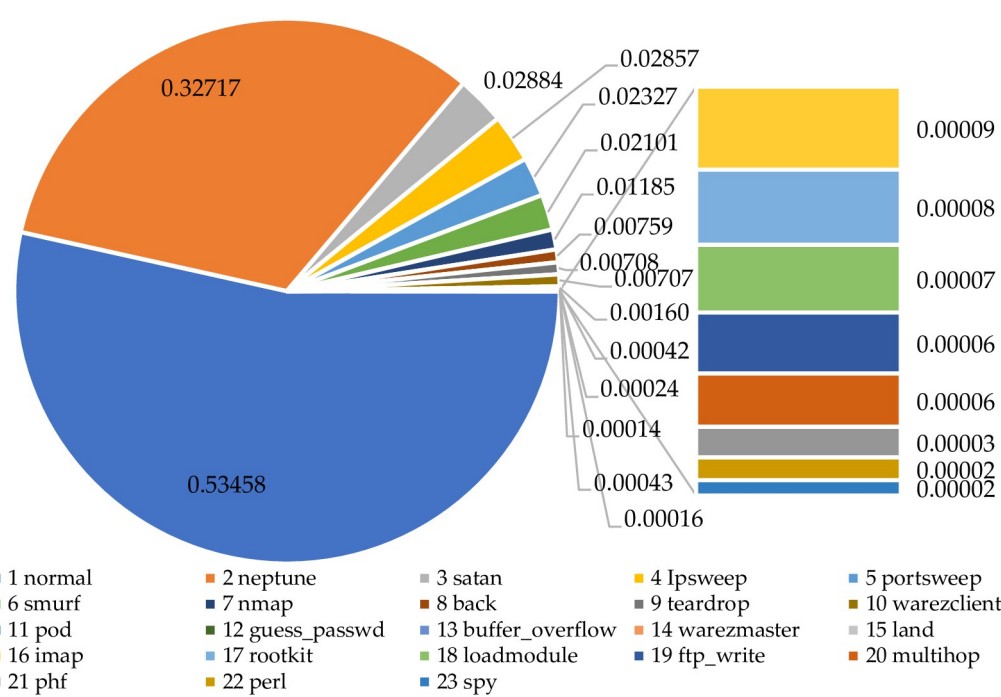

**Fig 5. Analysis chart of the proportion of all original data quantities.**

**Table 1. Classification results of NSL-KDD.**

| Serial Number | Label | Precision | Recall | F-value | Support |
|---|---|---|---|---|---|
| 1 | back | 1.00 | 1.00 | 1.00 | 95 |
| 2 | buffer_overflow | 1.00 | **0.60** | **0.75** | **5** |
| 3 | ftp_write | **0.00** | **0.00** | **0.00** | **0** |
| 4 | guess_passwd | 1.00 | 1.00 | 1.00 | 7 |
| 5 | imap | 1.00 | 1.00 | 1.00 | **1** |
| 6 | ipsweep | 0.99 | 1.00 | 1.00 | 388 |
| 7 | land | **0.33** | 1.00 | **0.50** | **1** |
| 8 | loadmodule | **0.00** | **0.00** | **0.00** | **0** |
| 9 | multihop | **0.00** | **0.00** | **0.00** | **0** |
| 10 | neptune | 1.00 | 1.00 | 1.00 | 4116 |
| 11 | nmap | 0.97 | 0.97 | 0.97 | 151 |
| 12 | normal | 1.00 | 1.00 | 1.00 | 6760 |
| 13 | perl | 1.00 | 1.00 | 1.00 | **1** |
| 14 | pod | 1.00 | 1.00 | 1.00 | 19 |
| 15 | **phf** | - | - | - | - |
| 16 | portsweep | 1.00 | 0.99 | 0.99 | 258 |
| 17 | rootkit | **0.00** | **0.00** | **0.00** | 1 |
| 18 | satan | 0.99 | 0.99 | 0.99 | 353 |
| 19 | smurf | 1.00 | 1.00 | 1.00 | 264 |
| 20 | spy | **0.00** | **0.00** | **0.00** | 1 |
| 21 | teardrop | 1.00 | 1.00 | 1.00 | 87 |
| 22 | warezclient | 1.00 | 0.98 | 0.99 | 89 |
| 23 | warezmaster | **0.50** | 1.00 | **0.67** | **1** |
| | macroavg | **0.72** | **0.75** | **0.72** | 12598 |
| | weightedavg | 1.00 | 1.00 | 1.00 | 12598 |

## 3.2. Solution strategy of imbalanced data of intrusion detection based on cGAN

Based on the above comprehensive analyses of the imbalanced data and the generation of the adversarial network, we propose a cGAN-based solution to the imbalance of intrusion detection data, as shown in Fig 6, to address the problem of missing data, coverage categories and low recognition rate of scarce data caused by the imbalance of attack types and quantities in network intrusion detection.

This strategy mainly includes four parts, namely generator module G, discriminator module D, target network F and identification network $\Phi_{\text{Classification}}$. The condition vector $R_{\text{label}}$ is flattened with Flatten layer and Embedding layer. Multiply layer is used to combine random noise $R_{\text{seed}}$ and $R_{\text{label}}$, which are converted to the input data shape acceptable to generator G. Generator G is responsible for learning the distribution rules of attack behavior and normal behavior data, extracting the probability distribution characteristics, and then generating an expanded attack sample that approximates the distribution of attack behavior data. At the same time, G receives feedback from the loss function and outputs $G_{\text{data}}$ and its corresponding label information $G_{\text{label}}$ that fits the potential spatial distribution pattern of the input data. The generator tries to "trick" the discriminator by generating samples so that it cannot correctly distinguish between real data samples and generated data samples. The target network F is composed of intrusion data. This paper uses the NSL-KDD dataset. In the F network, attacks

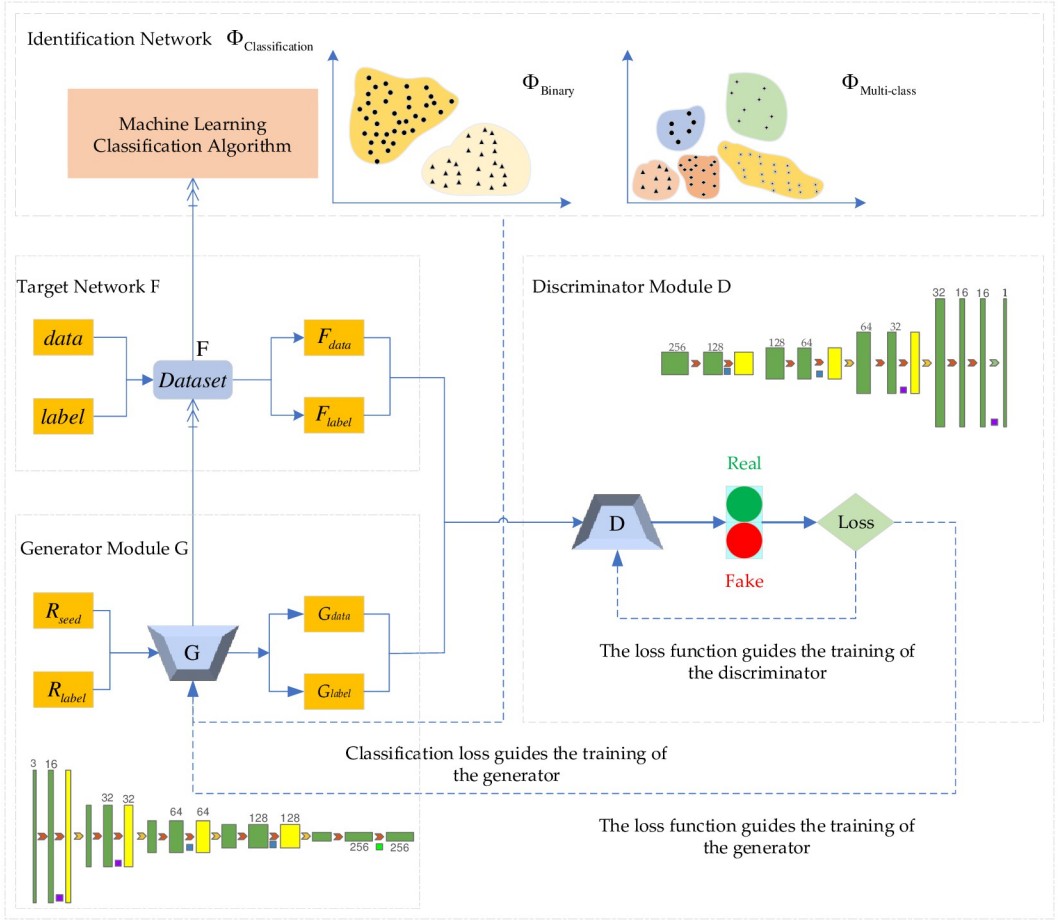

**Fig 6. Solution strategy of imbalanced data of intrusion detection based on cGAN.**

are unevenly distributed. After numericalization, data standardization, and normalization, $F_{data}$ and $F_{label}$ are generated to satisfy the shape of discriminator input data. The number of $F_{data}$ and $F_{label}$ is small in the total network traffic. The dimension will increase with the degree of use of the network. The variety will increase with the diversification of attack methods and attack tools. $F_{data}$ and $F_{label}$ tend to be ignored in the normal network intrusion detection system, which are the blind spots of the defense identification system. The discriminator D receives the generated sample $G_{data}$, the generated sample label information $G_{label}$, the real data attack behavior $F_{data}$, and the real data attack behavior data label $F_{label}$, and performs the judgment in the direction favorable to minimizing the loss. The discriminator D expects to discriminate the source of the input information, i.e., effectively distinguish $G_{data}$ and $F_{data}$. The discriminator result feeds optimization information through $G_{label}$ and $F_{label}$, which in turn improves the generator generation effect, fine-tunes the generation direction, and optimizes the fitting effect. The discriminator will use the computational advantages of deep learning in data processing to continuously reduce its loss, improve the model's comprehensive performance, and improve the generalization ability. The generator and the discriminator will improve their own generation effect and discriminative ability through loss change respectively, and finally reach the Nash equilibrium state.

## 3.3. The selection of attack samples

The flowchart of the attack algorithm is shown in Fig 7. F network consists of the data with characteristics of attack behavior that attack the network, forming an attack sample. One of the attack behavior data of the target network is selected from the set of attack algorithm, until the number of attack samples size equal to num.data is network traffic data, including normal behavior and attack behavior. label is the normal behavior and attack behavior category information. $F_{data}$ is the generated data by the model that matches the distribution characteristics of the attack behavior in the real traffic of the network. $F_{label}$ is the corresponding attack behavior category information, and num is a fixed value that can satisfy the classification index of the network attack behavior based on experience.

# 4. Results and analyses

## 4.1. Experimental environment and parameter settings

The computer system is Windows 10. The processor is Intel(R) Core (TM) i9-10920X CPU@3.50GHz with a running memory (RAM) of 128GB and dual NVIDIA Geforce RTX 3090 GPUs. The computer is equipped with python 3.7 and Tensorflow2.2 framework.

In the training process, the generator and discriminator of cGAN adopt a combination of convolutional network and Batch Normalization network unit structure as the generator and discriminator architecture. In the generator and discriminator, a "small and deep"

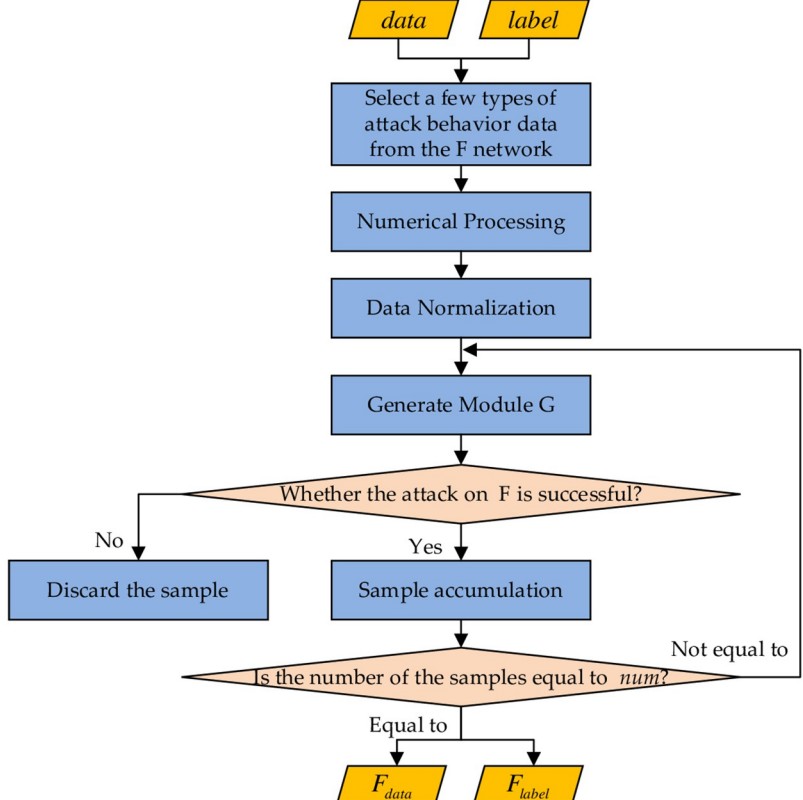

**Fig 7. The screening process of attack samples.**

convolutional structure is used to reduce the size of the convolutional kernel and increase the depth of the model. The generator uses the Tanh activation function except for the last layer, and other layers use the LeakyReLU activation function. The discriminator uses Sigmoid as the activation function. The normalization layer will normalize the input of the hidden layer.

The hyperparameters are set as follows.

1. noise_dim is set to 50;

2. num is set to 900;

3. Training batch EPOCHS is set to 800;

4. Batch size BATCH_SIZE is set to 256;

5. The learning rate of Adam optimizer is set to $1e - 5$.

In the training process, the generator and discriminator are trained alternately using a batch training method, with the batch set to EPOCHS and the batch size set to BATCH_SIZE. When training the generator, the parameters of the generator network are set empirically, and the random noise z and the conditional vector c of size noise_dim are obtained, where noise_-dim is the artificially set random noise dimension information. The random noise z and the conditional vector c are spliced and input into the generator to generate samples with the same dimension as the attack behavior features, and passed into the discriminator and classifier. The Adam optimizer is used to optimize the loss function of the generator, back-propagate and update the parameters of the generator. This step is repeated several times until the parameters cannot be further optimized. When training the discriminator, the discriminator network parameters are set empirically, the noise_dim size data are obtained from the training samples and input to the discriminator to optimize the corresponding loss function. The noise_dim size noise z and the conditional vector c are obtained and input to the generator to generate the generated samples with the same dimensionality as the attack behavior features. The loss function is back-propagated and the parameters of the discriminator are updated. The procedure is repeated until the parameters cannot be further optimized. Repeating the alternating training of the generator and discriminator until the network training is completed so that the valid attack behavior data can be "faked". Each complete cycle of training samples completes one round of training, and the relevant parameters of the generator and discriminator are saved.

## 4.2. Experimental results and analysis of data of minority class

As can be seen from Fig 8, after about 500 epochs, the model reaches game stability and achieves the corresponding dynamic balance between the generator and the discriminator, at which time the generative adversarial network structure has reached the best training effect. In principle, at this time, the generated samples of the generator can no longer be quickly and accurately identified by the discriminator. The discriminator has sufficient discriminatory ability, i.e., the distribution of the generated samples at this time is close to the original data samples. The generator part of the model is saved separately. Data of imbalanced distribution can be called at any time and generated to expand the operation to make up for the unbalanced data, thus ensuring the effectiveness of the sample and detection efficiency.

In order to carry out unbalanced sample augmentation, the number of generated samples is set at 900 groups. 900 groups are randomly selected from the original dataset, and a new dataset is formed by using the ratio of attack behavior data: normal behavior data approximately equal to 1:1 for binary classification validation of the decision tree. Taking the

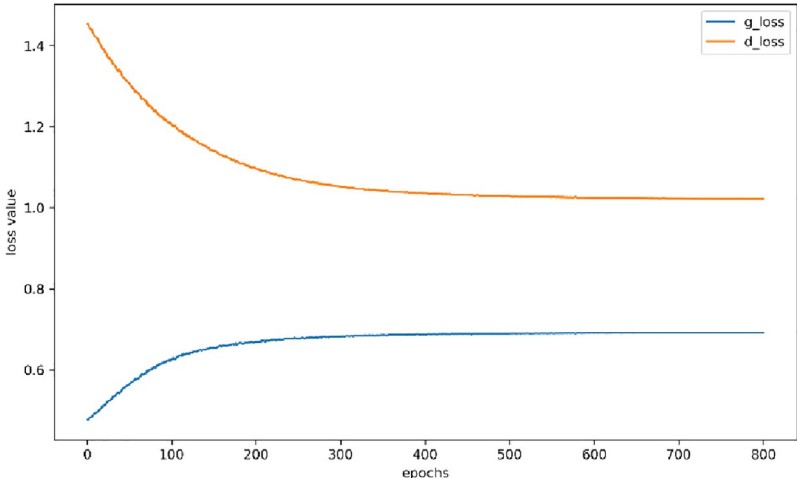

**Fig 8. Variation of the loss function of generator and discriminator of cGAN.**

buffer_overflow data as an example, the validation results are shown in Table 2. The accuracy can reach 99.46% at minimum. The precision rate is 0.99. The recall rate is 1.00. The F value is 0.99. The support degree is 463. The indicators show that the generated samples have the distribution characteristics of real attack behaviors, which is sufficient to replace the real samples for model training.

After comparing the accuracy, recall, F-value and support values of the data set classified by the decision tree algorithm before and after using the model proposed, as shown in Table 3, the experimental results of classification validity proved that the accuracy value changed from 1.00 to 0.99, the recall value changed from 0.60 to 1.00, the F-value changed from 0.75 to 0.99, and the support changed from 1 to 463. The decrease in the accuracy rate is due to the increase in the number of samples of a few classes, which "dilutes" the probability of the original data masking the attack data. The change in other data is extremely obvious.

In this paper, KFold (n_splits = 10) cross-validation method is used and the accuracy validity of the generated sample cases is verified using the feature selection evaluation experiment. The results are shown in Fig 9.

As can be seen from the Fig 9, the results of the cross-validation feature selection evaluation test indicate that the classification results are above 99.65% accuracy, indicating that the generated adversarial sample distribution fits the real attack behavior distribution more than expected. It can be concluded that the cGAN-based intrusion detection strategy can effectively solve the problem of unbalanced data and improve the intrusion detection effect.

**Table 2. Binary validation results after buffer_overflow data augmentation.**

|  | Precision | Recall | F-value | Support |
|---|---|---|---|---|
| normal | 1. 00 | 0. 99 | 0. 99 | 467 |
| buffer_overflow | 0. 99 | 1. 00 | 0. 99 | 463 |
| accuracy |  |  | 0. 99 | 930 |
| macro avg | 0. 99 | 0. 99 | 0. 99 | 930 |
| weighted avg | 0. 99 | 0. 99 | 0. 99 | 930 |

**Table 3. Comparison of binary classification results before and after using the model proposed for buffer_overflow data.**

|  | Precision | Recall | F-value | Support |
|---|---|---|---|---|
| Ours | 0. 99 | **1. 00** | **0. 99** | 463 |
| None | 1. 00 | **0. 60** | **0. 75** | **5** |

## 4.3. Experimental results and analysis of 23 classification based on NSL-KDD dataset

First, guess_passwd, buffer_overflow, warezmaster, land, imap, rootkit, loadmodule, ftp_write, multihop, phf, perl, and spy, a total of 12 attacks, are used respectively using the cGAN-based network intrusion. The data distribution after the augmentation is shown in Table 4. From the change of the augmentation ratio, we can see that the more serious the unbalanced distribution in the original data set is, the larger the augmentation ratio is, which indicates that the strategy proposed has research significance for this kind of problem.

Finally, we use the decision tree for 23 classifications to verify the effectiveness of the strategy proposed, and compare the data without the strategy in this paper. The results are shown in Table 5.

Table 5 shows that the precision rate, recall rate, F-value and support rate in the identification results of unbalanced intrusion data after the optimization of cGAN-based intrusion detection strategy are much higher than the precision rate and recall rate of direct classification without processing. The macro average precision rate rises from 0.72 to 0.93, an increase of 29.17%. The macro average recall rate rises from 0.75 to 0.93, an increase of 24%. The macro average F-value rises from 0.72 to 0.93, an increase of 29.17%, and the macro average support rises from 12598 to 13713, an increase of 8.85%. Under the premise that the data is valid in the network dataset after the sample augmentation is generated, the cGAN-based intrusion detection strategy has significantly improved the classification of network attacks and can effectively process the unevenly distributed data to improve the defense capability of the system.

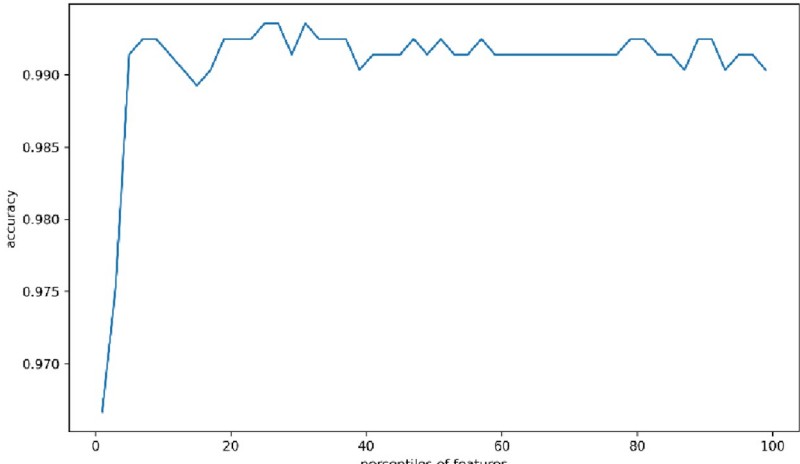

**Fig 9. Cross-validation feature selection evaluation results for buffer_overflow data.**

**Table 4. Unbalanced data widening.**

| Label | Before | After | Augmentation Ratio |
|---|---|---|---|
| guess_passwd | 53 | 1006 | 18.92 |
| buffer_overflow | 30 | 960 | 32 |
| warezmaster | 20 | 940 | 48 |
| land | 18 | 936 | 52 |
| imap | 11 | 922 | 83.82 |
| rootkit | 10 | 920 | 92 |
| loadmodule | 9 | 918 | 102 |
| ftp_write | 8 | 916 | 114.5 |
| multihop | 7 | 914 | 130.57 |
| phf | 4 | 908 | 227 |
| perl | 3 | 906 | 302 |
| spy | 2 | 904 | 452 |

Fig 10 shows the evaluation results of cGAN-based network intrusion detection strategy for multi-classification cross-validation feature selection. It can be seen from the Figure that the highest accuracy tends to be 98.72%. Such a high accuracy can be achieved after acquiring only 68% of the data features, so the model has obvious advantages.

**Table 5. Comparison of multi-classification results before and after using the cGAN-based intrusion detection model.**

| Serial Number | Label | Preci-sion | Ours-Precision | Recall | Ours-Recall | F-value | Ours-F-value | Support | Ours-Support |
|---|---|---|---|---|---|---|---|---|---|
| 1 | back | 1.00 | 1.00 | 1.00 | 1.00 | 1.00 | 1.00 | 95 | 118 |
| 2 | buffer_overflow | **1.00** | **0.94** | **0.60** | **0.91** | **0.75** | **0.93** | **5** | **89** |
| 3 | ftp_write | **0.00** | **0.95** | **0.00** | **0.94** | **0.00** | **0.94** | **0** | **81** |
| 4 | guess_passwd | 1.00 | 0.93 | 1.00 | 0.97 | 1.00 | 0.95 | 7 | **90** |
| 5 | imap | 1.00 | 0.97 | 1.00 | 0.98 | 1.00 | 0.97 | **1** | **99** |
| 6 | ipsweep | 0.99 | 0.99 | 1.00 | 0.97 | 1.00 | 0.98 | 388 | 325 |
| 7 | land | **0.33** | **0.90** | 1.00 | 0.92 | **0.50** | **0.91** | **1** | **90** |
| 8 | loadmodule | **0.00** | **0.64** | **0.00** | **0.59** | **0.00** | **0.61** | **0** | **85** |
| 9 | multihop | **0.00** | **0.65** | **0.00** | **0.67** | **0.00** | **0.66** | **0** | **95** |
| 10 | neptune | 1.00 | 1.00 | 1.00 | 1.00 | 1.00 | 1.00 | 4116 | 4100 |
| 11 | nmap | 0.97 | 0.96 | 0.97 | 0.97 | 0.97 | 0.96 | 151 | 152 |
| 12 | normal | 1.00 | 1.00 | 1.00 | 1.00 | 1.00 | 1.00 | 6760 | 6772 |
| 13 | perl | 1.00 | 0.93 | 1.00 | 0.98 | 1.00 | 0.95 | **1** | **96** |
| 14 | **phf** | _ | **0.98** | _ | **0.88** | _ | **0.93** | _ | **99** |
| 15 | pod | 1.00 | 1.00 | 1.00 | 1.00 | 1.00 | 1.00 | 19 | 19 |
| 16 | portsweep | 1.00 | 0.99 | 0.99 | 0.99 | 0.99 | 0.99 | 258 | 307 |
| 17 | rootkit | **0.00** | **0.82** | **0.00** | **0.87** | **0.00** | **0.84** | **1** | **92** |
| 18 | satan | 0.99 | 0.97 | 0.99 | 0.99 | 0.99 | 0.98 | 353 | 340 |
| 19 | smurf | 1.00 | 0.99 | 1.00 | 0.99 | 1.00 | 0.99 | 264 | 277 |
| 20 | spy | **0.00** | **0.90** | **0.00** | **0.96** | **0.00** | **0.93** | **1** | **107** |
| 21 | teardrop | 1.00 | 1.00 | 1.00 | 1.00 | 1.00 | 1.00 | 87 | 95 |
| 22 | warezclient | 1.00 | 0.92 | 0.98 | 0.99 | 0.99 | 0.95 | 89 | 84 |
| 23 | warezmaster | **0.50** | **0.91** | 1.00 | 0.93 | **0.67** | **0.92** | **1** | **101** |
|  | accuracy |  |  |  |  | 1.00 | 0.99 | 12598 | 13713 |
|  | macroavg | 0.72 | 0.93 | 0.75 | 0.93 | 0.72 | 0.93 | 12598 | 13713 |
|  | weightedavg | 1.00 | 0.99 | 1.00 | 0.99 | 1.00 | 0.99 | 12598 | 13713 |

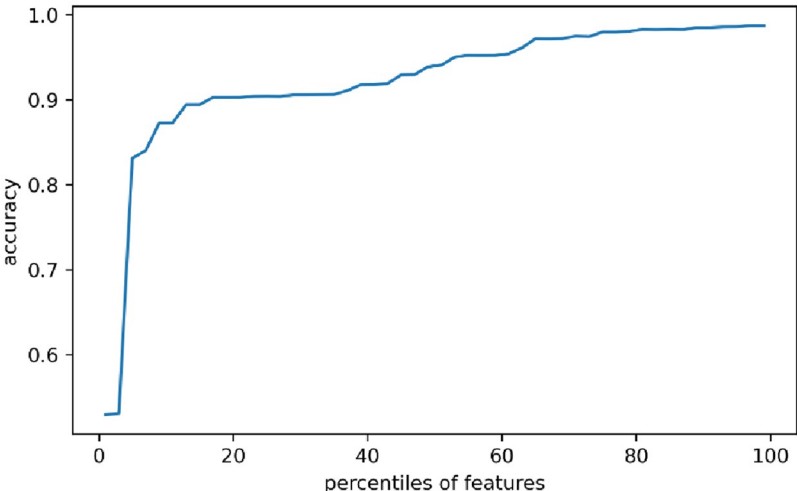

**Fig 10. Evaluation results of multi-classification cross-validation feature selection for cGAN-based network intrusion detection strategy.**

## 4.4. Comparison of experimental results of 5 classification of different intrusion detection methods based on NSL-KDD dataset

To further demonstrate the efficiency of the proposed intrusion detection strategy, existing literature results are selected and compared, mainly regarding accuracy, recall and F-value. These include BP [14], Radial Basis Function (RBF) [14], Regularized Extreme Learning Machine (RELM) [14], Improved Regularized Extreme Learning Machine (IRELM) [14], Genetic Algorithm-IRELM (GA-IRELM) [14], Particle Swarm Optimization-IRELM (PSO-IRELM) [14], Brain Storm Optimization-IRELM (BSO-IRELM) [14], Gradient Boosting Decision Tree (GDBT) [19], K-Nearest Neighbour (KNN) algorithm [19], GBDT-KNN [15], Restricted Boltzmann Machine (RBM) [16], Stacked Autoencoder (SAE) [16], Deep Belief Networks (DBN) [17], S-NDAE [16], Support Vector Machine (SVM) [18], Decision Tree J48 algorithm [18], Naive Bayesian Tree (NBTree) [18], Decision Tree (DT) [19], Naïve Bayesian algorithm [19], Expectation-Maximization clustering (EM clustering) [19], Artificial Neural Network (ANN) [19], Artificial Neuro Network, ANN [19], Random Forest (RF) [19], Adaptive Boosting (AdaBoost) [19]. The comparison results are shown in the following table.

For the classification experiments, accuracy is the clearest indicator of the performance of the strategies and models. As shown in Table 6, it can be clearly seen that the multi-classification accuracy of the proposed strategy reaches 98.65%, which is the highest among all the compared methods, and is at least 3.87% better compared to other algorithms.

In the presence of severe data categories and quantitatively unbalanced distribution, the accuracy rate can no longer fully represent the performance of the model, so other metrics need to be introduced. Therefore, in this paper, the accuracy rate, recall rate and F-value are used as progressive analysis metrics. The specific comparison results are shown in the following table and Figure.

The accuracy rate represents the relative correctness of the prediction results, i.e., the cases in which the samples predicted as attacks are really attacks, and is used to measure whether there are cases of misclassification. As can be seen from Table 7, regarding the precision rate results of the prediction of network behaviors in the classes Normal, Prob, Dos, U2R, and R2L, the scores of the strategy proposed are 1.000, 0.9775, 0.9817, 0.8325, and 0.9012, respectively.

**Table 6. Accuracy comparison of multi-classification results of different intrusion detection models.**

| Method | BP | RBF | RELM | IRELM | GA-IRELM |
|---|---|---|---|---|---|
| **Accuracy** | 0.7310 | 0.8195 | 0.6270 | 0.7190 | 0.8635 |
| **Method** | PSO-IRELM | BSO-IRELM | GBDT-KNN | RBM | SAE |
| **Accuracy** | 0.8615 | 0.8870 | 0.8959 | 0.7880 | 0.7930 |
| **Method** | DBN | S-NDAE | SVM | J48 | NBTree |
| **Accuracy** | 0.8058 | 0.8542 | 0.7400 | 0.7460 | 0.7540 |
| **Method** | DT | Naïve Bayesian | EM clustering | KNN | ANN |
| **Accuracy** | 0.9382 | 0.8166 | 0.8020 | 0.8432 | 0.9226 |
| **Method** | RF | GDBT | AdaBoost | **Ours** | |
| **Accuracy** | 0.9279 | 0.9456 | **0.9478** | **0.9865** | |

Among them, the precision rates for the classes Prob, Dos, U2R, and R2L are not the highest values, but the difference is small. The network intrusion attack behavior is not a single type of attack, so the overall performance of the model is required to be higher. Based on the accuracy rate values and the comprehensive data, the overall performance of the proposed strategy is much better than other intrusion detection methods.

The recall rate, which indicates the situation that a certain type of behavior in the dataset, is correctly predicted. Since the number of normal behaviors in the dataset is too high, this scenario will lead to a high accuracy of all model predictions but a weak identification of attack behaviors, which is not the desired result. At this point, the value of the recall rate comes into play to measure the ability to predict attack behaviors and check whether there are any omissions. As can be seen from Table 8, regarding the accuracy rate results of network behavior prediction in Normal, Prob, Dos, U2R, and R2L classes, the scores of the strategies in this paper are 1.0000, 0.9800, 0.9850, 0.8325, and 0.9150, respectively, with Normal, Prob, Dos, and R2L classes being the highest scores. Although the RELM algorithm scores a little higher in the U2R category, it has lower recall scores in the Normal, Dos and R2L categories. The overall performance of the proposed strategy is much better than other intrusion detection methods, based on the recall values and the comprehensive data.

The F-value is a criterion obtained by combining the precision and recall rates. The higher the F-value, i.e., the higher the summed average of precision and recall, the better the model.

**Table 7. Comparison of accuracy rates of multi-classification results of different intrusion detection models.**

| Method | Precision | | | | |
|---|---|---|---|---|---|
| | Normal | Prob | Dos | U2R | R2L |
| GBDT-KNN | 0.8534 | 0.9158 | 0.8895 | 0.9215 | 0.0836 |
| KNN | 0.6764 | 0.9210 | 0.8139 | **0.9762** | 0 |
| BP | 0.7778 | 0.8298 | 0.9302 | 0.5217 | 0.875 |
| RBF | 0.6694 | **0.9959** | **0.9941** | 0.6710 | 0.68 |
| AdaBoost | 0.5235 | 0.8964 | 0.9035 | 0.7363 | _ |
| SVM | 0.9129 | 0.7337 | 0.8471 | 0.8700 | _ |
| RELM | 0.8952 | 0.5177 | 0.9674 | 0.5900 | 0.500 |
| IRELM | 0.4976 | 0.7991 | 0.9671 | 0.6522 | 0.8182 |
| GA-IRELM | 0.8897 | 0.8989 | 0.9075 | 0.7676 | 0.9091 |
| PSO-IRELM | 0.8036 | 0.8976 | 0.9353 | 0.7772 | 0.9091 |
| BSO-IRELM | 0.8396 | 0.8966 | 0.9636 | 0.8095 | **0.9231** |
| Ours | **1.0000** | 0.9775 | 0.9817 | 0.8325 | 0.9012 |

**Table 8. Comparison of recall rates of multi-category results for different intrusion detection models.**

| Method | Recall | | | | |
|---|---|---|---|---|---|
| | Normal | Prob | Dos | U2R | R2L |
| GBDT-KNN | 0. 9722 | 0. 9563 | 0. 9767 | 0. 1389 | 0. 1235 |
| KNN | 0. 9557 | 0. 8684 | 0. 6216 | 0. 0131 | 0 |
| BP | 0. 0387 | 0. 9750 | 0. 8152 | 0. 9716 | 0. 3111 |
| RBF | 0. 7352 | 0. 5971 | 0. 9092 | 0. 9427 | 0. 5667 |
| AdaBoost | 0. 6684 | 0. 8247 | 0. 7064 | 0. 9267 | 0 |
| SVM | 0. 6304 | 0. 8878 | 0. 9013 | 0. 8857 | 0 |
| RELM | 0. 5341 | 0. 9770 | 0. 2507 | **0. 9747** | 0. 0345 |
| IRELM | 0. 5838 | 0. 9144 | 0. 5401 | 0. 9649 | 0. 3913 |
| GA-IRELM | 0. 6419 | 0. 9244 | 0. 8948 | 0. 9646 | 0. 5714 |
| PSO-IRELM | 0. 6131 | 0. 9448 | 0. 895 | 0. 9593 | 0. 5405 |
| BSO-IRELM | 0. 6618 | 0. 9388 | 0. 9361 | 0. 9404 | 0. 8276 |
| Ours | **1. 0000** | **0. 9800** | **0. 9850** | 0. 8325 | **0. 9150** |

From Table 9, it can be seen that the scores of our strategy are 1.0000, 0.9775, 0.9833, 0.8325, and 0.9063 for the accuracy rate results of network behavior prediction in the Normal, Prob, Dos, U2R, and R2L categories, respectively. The F-value of the R2L class is significantly less than the strategy proposed, and the comprehensive performance of the proposed strategy is much better than other intrusion detection methods based on the F-value as the criterion and the comprehensive data.

## 4.5 Comparison experiments on UNSW-NB15 dataset

In order to demonstrate the generalization ability of the method proposed, this paper is validated on the UNSW-NB15 dataset. The experimental setup and evaluation indexes are the same as those of the NSL-KDD dataset. The distribution of the number of attack categories in the UNSW-NB15 dataset is shown in Fig 11. Among them, the attack behavior data of Analysis, Backdoor, Shellcode and Worms categories account for 1.14%, 1.00%, 0.65% and 0.07% of the total number, respectively, which are highly unbalanced in terms of quantitative

**Table 9. Comparison of F-values of multi-classification results for different intrusion detection models.**

| Method | F-value | | | | |
|---|---|---|---|---|---|
| | Normal | Prob | Dos | U2R | R2L |
| GBDT-KNN | _ | _ | _ | _ | _ |
| KNN | _ | _ | _ | _ | _ |
| BP | 0. 0737 | 0. 8966 | 0. 8689 | 0. 6789 | 0. 4591 |
| RBF | 0. 7007 | 0. 7465 | 0. 9498 | 0. 784 | 0. 6182 |
| AdaBoost | 0. 5872 | 0. 8591 | 0. 7929 | 0. 8206 | _ |
| SVM | 0. 7458 | 0. 8035 | 0. 8734 | **0. 8778** | _ |
| RELM | 0. 6690 | 0. 6768 | 0. 3982 | 0. 7351 | 0. 0645 |
| IRELM | 0. 5373 | 0. 8529 | 0. 6931 | 0. 7783 | 0. 5294 |
| GA-IRELM | 0. 7458 | 0. 9114 | 0. 9011 | 0. 8549 | 0. 7018 |
| PSO-IRELM | 0. 6955 | 0. 9206 | 0. 9147 | 0. 8587 | 0. 6780 |
| BSO-IRELM | 0. 7401 | 0. 9172 | 0. 9497 | 0. 8701 | 0. 6909 |
| Ours | **1. 0000** | **0. 9775** | **0. 9833** | 0. 8325 | **0. 9063** |

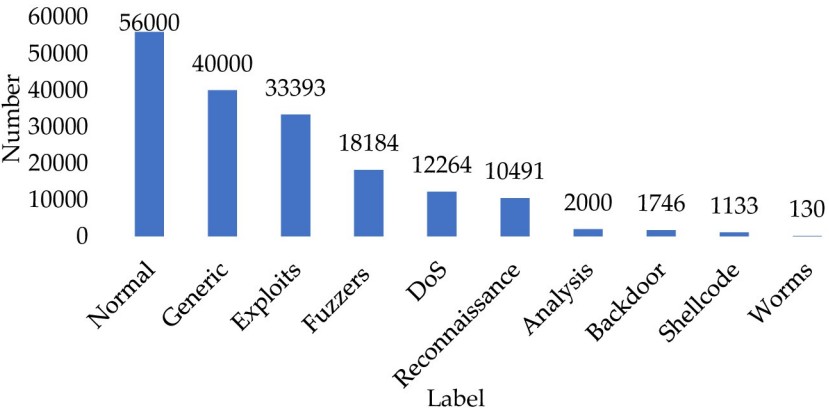

**Fig 11. Data distribution of UNSW-NB15 dataset.**

distribution compared to other types of attack data. Therefore, the experiments and comparisons are validated with Analysis, Backdoor, Shellcode and Worms types.

This part only aims to verify whether the method is valid in the UNSW-NB15 dataset and has sufficient generalization ability, so other hyperparameters are kept consistent with the NSL-KDD dataset experiments. The training batch EPOCHS is set to 2500.

To verify that the generated category data have sufficient characteristics to simulate the features of real category data, the binary classification validity experiments are conducted. The results are shown in Table 10 below, where the normal and attack attributes of the data can be efficiently discriminated. The generated data have the distribution characteristics of each type of attack data, which can solve the data omission and annexation problems caused by the unbalanced distribution.

To reflect the advanced nature of the proposed method, the detection performance is compared with 11 classification methods and 8 oversampling methods based on DNN. 11 classification methods are DT, Le-Net5 convolutional neural network, AlexNet convolutional neural network, Machine Learning based on convolutional network (ML-CNN), KNN, Multinomial Bayesian MultinomialNB, RF, SVM, ANN, DBN, and Deep Confidence Network-based Density Peak Clustering Algorithm-DBN (MDPCA-DBN) [23]. The oversampling methods are Random Over Sample (ROS) [20], SMOTE [6], ADASYN [8], SVMSMOTE [21], KMeansSMOTE [22], BorderlineSMOTE1 [7], BorderlineSMOTE2 [7], Improved Conditional Variational Autoencoder (Improved Conditional Variational Autoencoder, ICVAE) [23].

The comparison results are shown in Table 11 and Fig 12. In the Backdoor and Worms classes, our strategy achieves the highest detection rates of 0.82 and 0.92, and performs slightly

**Table 10. Experimental validation results of generation data validity of analysis, backdoor, DoS, reconnaissance, shellcode and worms class.**

| Label | Precision | Recall | F-value | Accuracy |
|---|---|---|---|---|
| Analysis | 1.00 | 1.00 | 1.00 | 1.0000 |
| Backdoor | 1.00 | 1.00 | 1.00 | 0.9957 |
| DoS | 0.99 | 1.00 | 1.00 | 0.9953 |
| Reconnaissance | 1.00 | 1.00 | 1.00 | 0.9979 |
| Shellcode | 0.99 | 1.00 | 0.99 | 0.9920 |
| Worms | 0.99 | 1.00 | 1.00 | 0.9961 |

**Table 11. Comparison of the detection performance of different classification methods and oversampling methods with the strategy proposed on the UNSW-NB15 dataset.**

| Label | DoS | Analysis | Backdoor | Worms |
|---|---|---|---|---|
| DT | 0.2500 | 0.5300 | 0.5400 | 0.1700 |
| Le-Net5 | 0.1621 | 0.1891 | 0.0703 | 0.0682 |
| AlexNet | 0.2345 | 0.1329 | 0.0189 | 0.0455 |
| ML-CNN | 0.1296 | 0.0236 | 0.0961 | 0.1591 |
| KNN | 0.1944 | 0.0148 | 0.0256 | 0.1111 |
| MultinomialNB | **0.7384** | 0.0000 | 0.0000 | 0.0000 |
| RF | 0.1760 | 0.0370 | 0.0513 | 0.1111 |
| SVM | 0.0000 | 0.0000 | 0.0000 | 0.0000 |
| ANN | 0.0513 | 0.0000 | 0.0000 | 0.0000 |
| DBN | 0.0232 | 0.0000 | 0.0000 | 0.0000 |
| MDPCA-DBN | 0.2372 | 0.0000 | 0.085 | 0.1111 |
| ROS | 0.1000 | 0.1314 | 0.4920 | 0.3409 |
| SMOTE | 0.2265 | 0.1536 | 0.4220 | 0.5227 |
| ADASYN | 0.0252 | **0.8656** | 0.0086 | 0.4773 |
| SVMSMOTE | 0.1978 | 0.0945 | 0.3379 | 0.3864 |
| KMeansSMOTE | 0.1558 | 0.4357 | 0.2041 | 0.3864 |
| BorderlineSMOTE1 | 0.1925 | 0.0399 | 0.8148 | 0.3409 |
| BorderlineSMOTE2 | 0.2675 | 0.0945 | 0.2676 | 0.2955 |
| ICVAE | 0.0792 | 0.1521 | 0.2058 | 0.7955 |
| Ours | 0.6000 | 0.8100 | **0.8200** | **0.9200** |

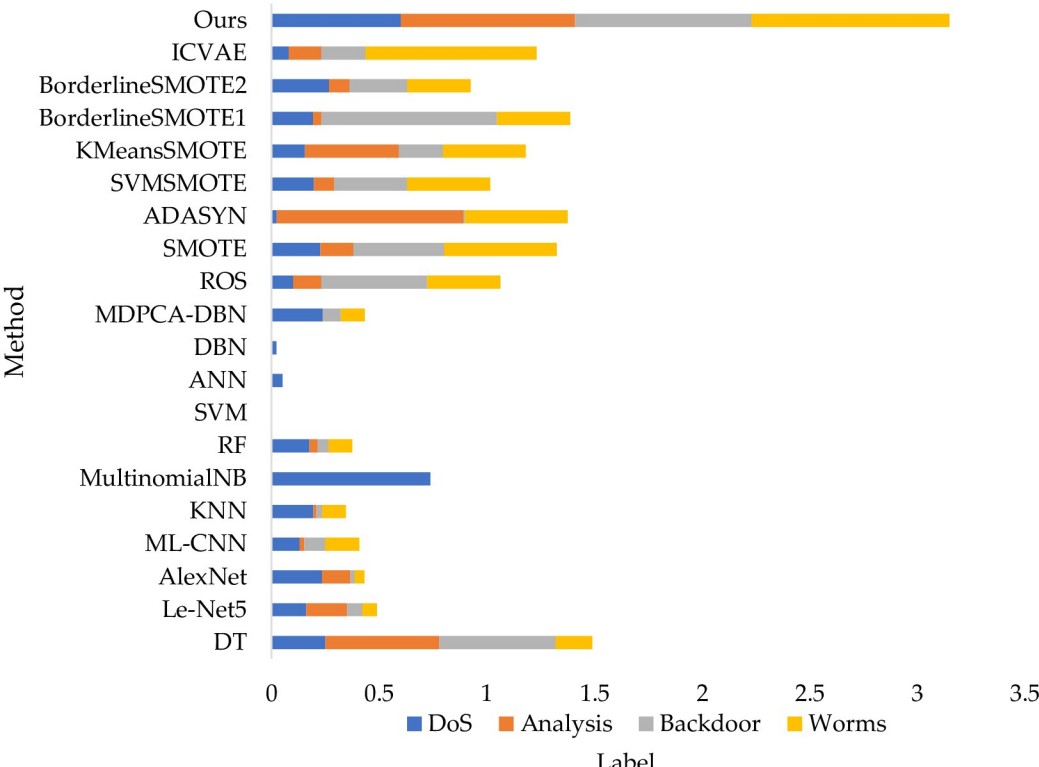

**Fig 12. Comparison of the detection performance of different classification methods and oversampling methods with the strategy of this paper on the UNSW-NB15 dataset.**

lower than MultinomialNB and ADASYN in the DoS and Analysis classes. However, MultinomialNB and ADASYN perform worse in the remaining classes. Combining the final detection rates of the four classes of data, we can see that the strategy presented has the best detection performance index.

## 5. Conclusions

In this paper, an intrusion detection method based on conditional generative adversarial networks is proposed. Taking advantage of the generative adversarial network model in data representation and distribution learning, the few classes of attack behavior data that are difficult to identify are augmented to ensure a random distribution within a certain bounded interval under the premise of having the potential spatial distribution pattern of real attack data. Based on the proposed strategy, the complex loss functions required to design existing deep learning methods are avoided by adding pooling layers, reducing the size of convolutional kernels, deepening the network structure, and adopting a batch training approach. The experimental results show that the proposed strategy presented can efficiently solve the problem of unbalanced data category and quantity share of attack behaviors in real network data streams, enhance the robustness of the classifier when facing different types of attack input data, improve the effectiveness of the intrusion detection model when identifying unbalanced data distribution, and enhance the generalization ability of the defense model to attack behaviors from different data sources.

## Supporting information

**S1 Data.**
(ZIP)

## Author Contributions

**Conceptualization:** Han Yang.

**Data curation:** Peng Liu, Ke Sun, Yang Yang.

**Formal analysis:** Peng Liu, Yang Yang, Tianyu Lan, Han Yang.

**Investigation:** Tianyu Lan, Han Yang.

**Project administration:** Yang Yang.

**Software:** Guangyu Zhao.

**Supervision:** Ke Sun, Han Yang.

**Validation:** Guangyu Zhao, Yang Yang, Tianyu Lan.

**Visualization:** Peng Liu.

**Writing – original draft:** Guangyu Zhao.

**Writing – review & editing:** Yang Yang.

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
