## [Decision Letter · Decision Letter 0]

19 Jun 2023

PONE-D-23-12856Research on Data Imbalance in Intrusion Detection Using CGANPLOS ONE

Dear Dr. Yang,

Thank you for submitting your manuscript to PLOS ONE. After careful consideration, we feel that it has merit but does not fully meet PLOS ONE’s publication criteria as it currently stands. Therefore, we invite you to submit a revised version of the manuscript that addresses the points raised during the review process.

We look forward to receiving your revised manuscript.

Kind regards,

Furqan Rustam

Academic Editor

PLOS ONE

“This research was funded by the National Natural Science Foundation of China, grant number U2141231.

the Science and Technology Development Program of Jilin Province, grant number 20200401066GX.

The Science and Technology Development Plan Project of Jilin Province, grant number 20200404216YY.”

Additional Editor Comments:

Dear Authors,

Reviewer comments suggest some suggestions to improve the quality of the manuscript please work on them and resubmit your revised article.

Reviewers' comments:

Reviewer's Responses to Questions

**Comments to the Author**

1. Is the manuscript technically sound, and do the data support the conclusions?

Reviewer #1: Yes

Reviewer #2: Yes

2. Has the statistical analysis been performed appropriately and rigorously? 

Reviewer #1: Yes

Reviewer #2: Yes

3. Have the authors made all data underlying the findings in their manuscript fully available?

Reviewer #1: Yes

Reviewer #2: Yes

4. Is the manuscript presented in an intelligible fashion and written in standard English?

Reviewer #1: Yes

Reviewer #2: Yes

5. Review Comments to the Author

Reviewer #1: the Manuscript "Analyzing influence of COVID-19 on crypto &financial markets and sentiment analysis

using deep ensemble model" is data rich research article with analytics and No further comments to make.

Reviewer #2: The proportion of various types of attack data to the total

37 traffic data is less than 0.1%. (citation needed)

Figure 4. shows the distribution of label data can you explain it is before augmentation or after . If it is before then you must include a grpah that shows balacned catagorical valaues.

Need to add detials of ML models that are used in this manuscript

6. PLOS authors have the option to publish the peer review history of their article (what does this mean?). If published, this will include your full peer review and any attached files.

Reviewer #1: **Yes: **Pradeep gali

Reviewer #2: No

---

## [Author Response · Author response to Decision Letter 0]

3 Sep 2023

Reviewer#2, Concern # 1: The proportion of various types of attack data to the total 37 traffic data is less than 0.1%. (citation needed).

Author response: We thank the reviewer for pointing this out and We have revised as follows.

Author action: We updated the manuscript by adding further detailed analysis results of the data. The specific changes are as follows:

Figure 5. Analysis chart of the proportion of all original data quantities

The analyses of the attack samples are shown in Figure 4. And combined with the proportion analysis of Figure 5, the distribution of categories is extremely unbalanced, with 12 categories accounting for about 0.1%, which are guess_passwd(0.042%), buffer_overflow(0.024%), warezmaster(0.016%), land(0.014%), imap(0.009%), rootkit(0.008%), loadmodule(0.007%), ftp_write(0.006%), multihop(0.006%), phf(0.003%), perl(0.002%) and spy(0.002%). The percentage of each type of data varies greatly. The number of unbalanced classes accounts for 52.17% of the total number of classes, and the amount of data only accounts for 1.39% of the total amount of data, which tend to mislead the classifier and cause detection omission and inaccurate identification. In addition, since the percentage is small, even if the abnormal behavior is recognized, the recognition rate is too low to attract attention.

Reviewer#2, Concern # 2: Figure 4. shows the distribution of label data can you explain it is before augmentation or after. If it is before then you must include a grpah that shows balanced catagorical valaues.

Author response: We feel sorry for the inconvenience brought to the reviewer. First, let's explain your first question. Figure 4. Shows the distribution of label data is before augmentation. Secondly, for the data before augmentation, we have added corresponding more detailed multi-classification results, macroavg and weightedavg, as shown in the revised Table 1. Moreover, in order to describe Figure 4. more clearly, we have revised the title of Figure 4, focusing on the distribution of the original unmodified data described in Figure 4. As shown in the modified Figure 4, it is the distribution of the original data. In Table 1, after modifying according to your suggestion, we can more clearly see the balanced classification results of the 23 types of data for subsequent experimental comparison.

Author action: We updated the manuscript by adding more detailed multi-classification results in Table 1, as follows:

Figure 4. Distribution of the original label data

Table 1. Classification results of NSL-KDD

Serial Number Label Precision Recall F-value Support

1 back 1.00 1.00 1.00 95

2 buffer_overflow 1.00 0.60 0.75 5

3 ftp_write 0.00 0.00 0.00 0

4 guess_passwd 1.00 1.00 1.00 7

5 imap 1.00 1.00 1.00 1

6 ipsweep 0.99 1.00 1.00 388

7 land 0.33 1.00 0.50 1

8 loadmodule 0.00 0.00 0.00 0

9 multihop 0.00 0.00 0.00 0

10 neptune 1.00 1.00 1.00 4116

11 nmap 0.97 0.97 0.97 151

12 normal 1.00 1.00 1.00 6760

13 perl 1.00 1.00 1.00 1

14 pod 1.00 1.00 1.00 19

15 phf - - - -

16 portsweep 1.00 0.99 0.99 258

17 rootkit 0.00 0.00 0.00 1

18 satan 0.99 0.99 0.99 353

19 smurf 1.00 1.00 1.00 264

20 spy 0.00 0.00 0.00 1

21 teardrop 1.00 1.00 1.00 87

22 warezclient 1.00 0.98 0.99 89

23 warezmaster 0.50 1.00 0.67 1

 macroavg 0.72 0.75 0.72 12598

 weightedavg 1.00 1.00 1.00 12598

To further verify the effect of unevenly distributed data on the classifier, the decision tree model is used to classify the features of the KDDTrain+.txt file. The classification results are shown in Table 1 with key information highlighted. It can be seen that the class distribution is extremely uneven, resulting in unsatisfactory categories of accuracy, recall, F-value and support, or even being 0, which does not have any reference value. The detection results do not show phf category attack behavior, which is a missed detection behavior, meaning a reduced reference value and effectiveness of the classification accuracy. In addition, in the multi-classification results of these 23 types of data, the macroavg value of precision is 0.72, the macroavg value of recall is 0.75, and the macroavg value of F-value is 0.72. It can be seen that the result is very unsatisfactory. In real network intrusion attacks, the distribution of attack behavior compared to normal behavior and this situation is similar. Therefore, great attention is needed to the problem of unbalanced data distribution, which is also the focus of this paper.

Reviewer#2, Concern # 3: Need to add detials of ML models that are used in this manuscript

Author response: Thank you very much for your valuable comments in this regard.The ML model is indeed extremely important for the classification problem itself.For this article, it is normal for you to have questions in this regard.The main focus of this article is that in classification problems, there are often many categories of objects to be classified, but the data capacity of a single category of classified objects is very scarce.In this case, the distribution characteristics of category data with scarce capacity cannot be better extracted by existing classification methods, and even problems such as category duplication, vague data categories, and omissions in offensive behavior will arise.The ML model is only used to prove that the extremely unbalanced data has been processed by the method of this article, which has significantly improved the recognition rate and accuracy rate. Therefore, the choice of ML model is not the most important value of this article.Of course, if you are more interested in this aspect, we are very willing and very happy to discuss with you in more detail in this field.Thank you very much again for your review, and look forward to your contacting our private email address to discuss this aspect.

Author action: We have explained in detail in the author response.

---

## [Editor Report · Decision Letter 1]

5 Sep 2023

Research on Data Imbalance in Intrusion Detection Using CGAN

PONE-D-23-12856R1

Dear Dr. Yang,

We’re pleased to inform you that your manuscript has been judged scientifically suitable for publication and will be formally accepted for publication once it meets all outstanding technical requirements.

Kind regards,

Furqan Rustam

Academic Editor

PLOS ONE

Additional Editor Comments (optional):

Authors resolved all comment very well.
---

## [Editor Report · Acceptance letter]

2 Oct 2023

PONE-D-23-12856R1 

Research on Data Imbalance in Intrusion Detection Using CGAN 

Dear Dr. Liu:

I'm pleased to inform you that your manuscript has been deemed suitable for publication in PLOS ONE. Congratulations! Your manuscript is now with our production department. 

Kind regards, 

on behalf of

Mr. Furqan Rustam 

Academic Editor

PLOS ONE